# Advances in Plant Epigenome Editing Research and Its Application in Plants

**DOI:** 10.3390/ijms24043442

**Published:** 2023-02-08

**Authors:** Qiaoyun Qi, Bichun Hu, Weiyu Jiang, Yixiong Wang, Jinjiao Yan, Fengwang Ma, Qingmei Guan, Jidi Xu

**Affiliations:** 1State Key Laboratory of Crop Stress Biology for Arid Areas/Shaanxi Key Laboratory of Apple, College of Horticulture, Northwest A&F University, Xianyang 712100, China; 2College of Forestry, Northwest A&F University, Xianyang 712100, China

**Keywords:** genome editing, epistatic regulation, CRISPR/Cas9, plant

## Abstract

Plant epistatic regulation is the DNA methylation, non-coding RNA regulation, and histone modification of gene sequences without altering the genome sequence, thus regulating gene expression patterns and the growth process of plants to produce heritable changes. Epistatic regulation in plants can regulate plant responses to different environmental stresses, regulate fruit growth and development, etc. Genome editing can effectively improve plant genetic efficiency by targeting the design and efficient editing of genome-specific loci with specific nucleases, such as zinc finger nucleases (ZFNs), transcription activator-like effector nucleases (TALEN), and clustered regularly interspaced short palindromic repeats/CRISPR-associated 9 (CRISPR/Cas9). As research progresses, the CRISPR/Cas9 system has been widely used in crop breeding, gene expression, and epistatic modification due to its high editing efficiency and rapid translation of results. In this review, we summarize the recent progress of CRISPR/Cas9 in epigenome editing and look forward to the future development direction of this system in plant epigenetic modification to provide a reference for the application of CRISPR/Cas9 in genome editing.

## 1. Introduction

Most of the domestication improvements in crops originate from genetic mutations. Usually, the probability of natural crop variation is not high, and crop improvement often takes a long period time. With the development of breeding technology, artificial mutation breeding has been widely used. Mutation breeding introduces random mutations in the genome with physical or chemical reagents, resulting in obtaining the target phenotype; however, its random introduction of mutations leads to inconsistent results and low efficiency in obtaining superior traits. Genome editing is performed by sequence-specific nucleases for precise localization and precise insertion or cutting of specific sequence fragments at target locations in the crop genome to silence or homologously recombine genes, thus allowing the expression of the target trait. The interplay between whole genome sequencing and genome editing allows for precise manipulation of target genes and is an efficient way to study mutant plants and create new germplasm. Genome editing technology is more straightforward in design, less costly, more efficient, and more operative than traditional breeding techniques, and it has emerged to reduce the difficulty of breeding and is now widely used in the plant field [1]. Genome editing technology has evolved rapidly through three technological innovations: zinc finger nucleases (ZFNs), transcription activator-like effector nucleases (TALEN), and clustered regularly interspaced short palindromic repeats/CRISPR-associated 9 (CRISPR/Cas9) systems.

Epigenetic modifications have essential effects on chromatin and gene regulation and play an important role in crop improvements, such as regulation of plant growth, resistance to environmental stresses, disease resistance, and regulation of fruit growth. Epigenetic changes, such as DNA methylation, histone modifications, and non-coding RNA changes, can affect genome stability and gene expression. Plants have a diverse epigenome, facilitating plant growth and development and enabling them to adapt to different environments. Plants have diverse epigenomes, and epigenetic editing at the genome level can further explore the relationship between epigenetic modifications and gene expression and facilitate plant adaptation to complex survival environments. Epigenetic modifications at the genome-wide level can further improve plant genetic efficiency and enhance the ability of plants to cope with different biotic and abiotic stresses and achieve precise regulation. By fusing the dCas9 protein with methylation- and histone-modifying enzymes, the expression of the target gene can be directly regulated by epigenetic modification of the gene [2]. These epigenomic alterations can effectively design target traits and also activate silenced genes to perform new functions in plant development. This paper reviews genome editing technologies, the types of epigenomes, and the successful applications of CRISPR/Cas9 in plant epigenomes.

## 2. Genome Editing Tools

With the continuous development of sequencing technology, whole genome sequencing has been completed in a variety of crops. Efficient modification of genomes is an effective way to facilitate plant research. The development of sequence-specific DNA nucleases, which allow genome editing technologies to target specific microsite genes for modification, has a wide range of developmental prospects. Currently, three specific nucleases are used for gene editing: zinc finger nuclease (ZFN), transcription activator-like effector nuclease (TALEN), and CRISPR/Cas. CRISPR/Cas system-mediated genome editing has significant applications in crop genetic improvement and molecular breeding due to its low cost, precision, and rapidity.

### 2.1. ZFN and TALEN

ZFN is one of the first genome editing technologies that emerged in the 1990s and is capable of precise mutations at specific sites [3]. It consists of a zinc finger protein (ZFP), used to recognize and bind specific DNA sequences, and the nuclease Fok I, which cuts DNA non-specifically. ZFN requires recognition as a dimer, a DNA-binding domain that mediates the binding of ZFN to the target gene-specific site, and the restriction of the endonuclease Fok I to dimerize and cut DNA to produce a double bond break, which in turn triggers the organism’s repair. This technique has a high target binding efficiency; however, the limited variety of zinc finger proteins, the high cost of zinc finger nuclease design, and the small number of DNA sequences that can be specifically recognized make genome editing less efficient, and its use has gradually decreased in recent years.

In 2009, the transcription activator-like effector nuclease (TALEN), based on ZFN, has more designability and consists of two parts: a cleavage domain and a binding domain with specific recognition. The cleavage domain is the restriction endonuclease FokI, and the binding domain consists of highly repetitive conserved sequences. These sequences are derived from the transcription activator-like effector (TALE) found in the phytopathogenic bacterium *Xanthomonas* spp. [4]. Moreover, it consists of 34 highly conserved amino acids, of which two variable amino acid residues at positions 12 and 13 are responsible for recognizing and binding DNA sequences that determine recognition specificity. The conserved structural domains bind specifically to target sites in the plant genome, forming a dimer that exerts endonuclease activity, leading to double-stranded DNA breaks in TALEN’s sequence spacer region, inducing a DNA damage repair mechanism and achieving the purpose of gene editing [5]. Compared with ZFN, the repetitive variable double amino acid residues in TALEN can recognize different bases, which is simpler to design and easier to screen. However, the technical construction of TALEN in the plant genome is more complicated, and the efficiency of genome editing is low and the cost is high.

### 2.2. CRISPR/Cas

The CRISPR/Cas system, widely found in bacteria and archaea, is an adaptive immune response protein mainly used to defend against invasion by exogenous DNA and viruses and was developed in 2012 [6]. By integrating the DNA fragments of invading exogenous genes into CRISPR sequences, crRNA (targeting CRISPR RNA) is transcribed and then combined with tracrRNA (transactivating CRISPR RNA) to form guide RNA (single guide RNA, sgRNA). Finally, sgRNA binds to Cas nuclease to degrade homologous complementary sequences to counteract exogenous genes such as phages [7]. With modification of the CRISPR system, it is easy to design and change the sgRNA sequence to almost any specific target sequence in the genome. It has become a common tool for crop improvement and genetic transformation.

Among the three types of CRISPR/Cas systems (Type I, Type II, and Type III), Type II system CRISPR/Cas9 requires only one Cas9 protein for precise cleavage of double-stranded DNA target sites, making in vitro design and modification easier to operate [8]. In contrast, the Type I and Type III CRISPR/Cas systems require multiple Cas proteins to form a nuclease complex to cleave DNA in a double-stranded manner, with a more complex mechanism of action. CRISPR/Cas9 is currently the most widely used system for genome editing due to its simplicity and efficiency. The CRISPR/Cas9 system includes the Cas9 protein and sgRNA (single-guide RNA). sgRNA is created by chimerizing the 5′-end of tracrRNA with the 3′-end of CRISPR RNA (crRNA), and Cas9 is created by hybridizing the 5′-end of tracrRNA with the 3′-end of CRISPR RNA (crRNA). The protein is composed of two parts: the NUC nuclease structural domain and the REC structural domain, which in turn contains the RuvC structural domain, the PAM structural domain, the HNH structural domain, and the WED structural domain. About 20 bases at the 5′ end of sgRNA determine the specific recognition of the DNA sequence. Then, the ribonucleoprotein complex is formed by the co-binding of sgRNA and Cas9 protein (the ribonucleoprotein complex (RNP), which binds to the target sequence, performs cleavage and causes double-strand breaks, triggering an endogenous mechanism to stimulate DNA repair for gene-editing purposes [9]. The genome of an organism theoretically produces one editable PAM region in every eight bases. Thus, the CRISPR/Cas9 system can edit genes for almost all organisms.

As the CRISPR/Cas9 system progresses, in addition to being used to edit target DNA or RNA, Cas9’s ability to integrate and function with effector proteins as a single subunit nuclease will significantly expand its applications in gene expression regulation, single-base editing, epigenetic modification, and imaging of living plant cells. Cas9 was transformed into dCas9 with no catalytic activity so the RNP complex could only recognize the target sequence without inducing a double-strand break, and the effector protein was fused with dCas9 [10]. This fusion protein is needed to form a dimer structure to target a specific gene’s promoter or enhancer region under the guidance of sgRNA. dCas9 is less likely to miss the target, improving the specificity of gene editing. Because dCas9 does not cut sequences and does not introduce mutations into the original genome, dCas9 can directly regulate gene expression, regulate plant growth, and produce heritable phenotype changes through epigenetic modification of the target genome through fusion with modifying enzymes. Therefore, CRISPR/ dCas9 has broad application prospects in plant epigenetic modification.

## 3. Types of Epigenetic Modifications of Genomes

Unlike conventional plant genome editing, epigenetic editing does not cut DNA and does not require repair. It affects gene expression and regulates plant growth and genetic transformation by targeting chemical and protein tags located on chromatin for reversible modifications with the help of modifying enzymes without altering the genome sequence. The environment influences phenotypic changes caused by epigenetic modifications, and both biotic and abiotic factors may cause changes in epigenetic modifications. There are close links between individual modifications, which are regulated by complex mechanisms. This allows plants to regulate their response to environmental stress finely and to inherit this response with some degree of stability. Epigenetic modification of DNA, histones and other components involved in gene expression can induce stable phenotypic changes inherited in mitosis or meiosis. In recent years, the relationship between plant life activities and epigenetic modification has been gradually discovered, revealing the complex interaction mechanism between environmental factors and plants. Plants have a diverse epigenome which facilitates plant growth and enhances the ability of plants to adapt to different environments [11]. Epigenomic modifications of the genome mainly include DNA methylation, histone modifications, and non-coding RNA actions, which play an important role in the regulation of gene expression.

### 3.1. DNA Methylation

DNA methylation is the most widely studied type of epigenetic modification that regulates gene expression by altering chromatin structure, DNA conformation, DNA stability, and DNA–protein interactions and also maintains genome stability by repressing transposons and transcription of exogenous DNA. In plants, DNA methylation is a necessary process that regulates gene expression and silences TEs and repetitive sequences. For the plant body, environmental changes lead to corresponding epigenetic variations. DNA methylation performs important functions in plant response to environmental stress, flowering regulation, and somatic asexual line development [12]. With the development of genome-wide methylation sequencing technology, the location of DNA methylation and its relationship with gene regulation can be determined on a genome-wide scale.

DNA methylation is an important feature of plant epigenomes and is involved in the formation of heterochromatin, affecting gene expression, transposon silencing, histone modifications, and many other aspects. Commonly, DNA methylation occurs at the 5th carbon atom of cytosine (C) on the DNA strand, forming 5-methylcytosine (5mC), with three different types including CG, CHG, and CHH (H being base A, T, or C). In addition, a small proportion of adenine (A) in plants can introduce a methyl group at the N6 position to form N6-methyladenine (N6-mA). Of these, 5-mC is mainly enriched around enhancer and promoter sites, an indication of repressed gene transcription, while N6-mA has a high abundance at the transcription start site, a sign of high gene transcriptional activity [13]. In plants, methylation can occur in symmetrical CG and CHG sequences, catalyzed by MET1 and CMT2/3, respectively, and modified at the corresponding positions in the nascent strand replicated by DNA semi-conservation, i.e., maintenance methylation. In contrast, CHH is methylated by CMT2 and DRM1/2 through the RNA-mediated DNA methylation (RdDM) pathway and relies on maintenance methylation enzymes to maintain its steady state, i.e., de novo methylation [14].

Specific methylation enzymes maintain different types of DNA methylation, e.g., MET1 (methyltransferase 1) is responsible for maintaining CG methylation, CMT3 (chromomethylase 3) is required for maintaining CHG methylation, DRM2 (domains rearranged methylase) is responsible for maintaining CG methylation and DRM2 (chromomethylase 3) is required for maintaining CHG methylation. Rearranged methylase 2 is primarily responsible for maintaining methylation at the CHH site. It also plays a role in maintaining DNA methylation [12].

In plants, DNA methylation mainly occurs at transposable elements (TEs) and repetitive DNA sequences. Transposons are distributed in the genomes of most eukaryotes but are unstable within the genome and may interfere with normal DNA sequences, and transposons vary greatly between species. Methylation and transposon activity are closely related; methylation is relatively stable and can be stably present during DNA replication, and plants usually methylate transposons, especially CHG and CHH, to suppress transposon activity [14]. Thus, DNA methylation plays a vital role in stabilizing the genome and participating in host defense. However, in contrast, some active transposons and genomic instability also benefit plants under certain conditions, such as DNA methylation-directed transposon insertions that sometimes give rise to new genes and play a positive role in plant evolution. Genome-wide methylation levels in plants are dynamically regulated by DNA methylation and active demethylation (ADM). Active demethylation is the active removal of methyl groups catalyzed by DNA demethylases [15].

Genome-wide methylation levels in plants are dynamically regulated by DNA methylation and active demethylation [16]. In the Arabidopsis DNA methylation and active demethylation regulatory system, the DNA demethylase gene ROS1 has a DNA methylation monitoring sequence (MMS) in its promoter, and the RdDM pathway increases genome-wide methylation levels, including MMS, when high levels of MEMS can activate ROSI expression; ROS1, a DNA demethylation enzyme, begins to perform demethylation functions and reduces genome-wide DNA methylation levels, including MEMS methylation levels. This coordinated mechanism of methylation and active demethylation can prevent excessive methylation levels, thus achieving a dynamic balance [17].

DNA methylation effectively regulates plant growth and development as well as responses to different environmental stresses. It can regulate gene expression at different levels of DNA methylation in different stages and tissues of plant growth and development, leading to changes in plant phenotype [16].

In response to changes in environmental factors, plants maintain their stability through reversible changes in DNA methylation and demethylation. A typical example is vernalization in plants, where prolonged low-temperature treatment produces a pronounced vernalization response in which low temperature induces the demethylation of flowering-related genes [18]. *Flowering Locus C* (*FLC*) is a key gene in vernalization and DNA methylation modifications negatively regulate its activity during vernalization. The *FLC* gene is expressed during the nutrient growth period, thereby inhibiting the development of reproductive growth. Under low-temperature stress, the activatory histone modifications H3K4me3 and HAC were replaced by the inhibitory modifications H3K9me3 and H3K27me3, suppressing FLC for *SOC1* (*overexpression of Co1*), *FT* (*flowering locus T*), and *FD* (*flowering locus D*) to promote plant reproductive growth [19]. Another typical example of the relationship between DNA methylation and abiotic stresses is the response of plants to drought. Under drought stress in plants, drought tolerance traits are primarily associated with reduced DNA methylation levels. Drought can induce changes in DNA methylation in rice and initiate the expression of drought-related genes, resulting in enhanced drought tolerance in rice, and such low methylation levels also have an effect on yield, with a negative correlation between single-plant yield and methylation levels [20]. About one-fourth of these drought-induced methylation changes could not be restored to the original state after rehydration, and about half of the methylated sites could not be restored and could be inherited by the next generation [21].

### 3.2. Histone Modifications

Similar to DNA methylation, histone modification is also an important form of epigenetic modification. Nucleosomes are the basic units of chromosomes and consist of a core octamer of four histone dimers, H2A, H2B, H3, and H4, bound to DNA sequences. Some covalent modifications occur on these four histones, and common histone modifications include methylation, acetylation, ubiquitination, phosphorylation, and glycosylation. These modifications promote or repress the expression of related genes by affecting how tightly histones bind to DNA or to other proteins, altering the dense or loose state of chromatin [22]. These histone post-translational modifications can regulate the transcriptional activity of genes by altering chromatin structure or recruiting downstream effectors. In plants, histone methylation and acetylation functions are better understood. Histone methylation sites are mainly localized on the lysine and arginine residues of H3 and H4, where three different states of modification can occur on histone lysine residues, monomethylated (me1), dimethylated (me2), and trimethylated (me3) [23]. The dynamic regulation of histone methylation relies on the combined action of histone methyltransferases (HMTs), histone methylation recognition proteins, and histone demethylases. The most widely studied modification sites include H3K9, H3K27, H3K4, H3K36, and other lysine methylation modifications on H3 histones. Usually, dimethylation and trimethylation modifications to H3K4 and H3K36 are associated with the activation of related genes, and methylation modifications to H3K9 and H3K27 are usually associated with the repression of genes, where H3K27me3 is highly reversible and plays an important dynamic regulatory role in plant development [24]. Acetylation and phosphorylation are the more active regulatory sites and tend to be the more stable modifications than methylation. H3K4 and H3K36 are usually considered active marks and represent active transcriptional regions in chromatin; H3K9 and H3K27 methylation are usually repressive marks and represent gene-silencing regions in chromatin. Similar to histone methylation, histone acetylation is regulated by a dynamic balance between histone acetyltransferases (HATs) and histone deacetylases (HDACs) [25].

There is an extensive linkage between histone modifications and DNA methylation, and effective silencing of transposons also requires coordination between DNA methylation and histone modifications, with H3K9 and H3K27 methylation being two hallmarks of transcriptional silencing in plants. For example, Arabidopsis H3K4me1 is associated with CG methylation of DNA [26]. Rice subjected to flooding stress induce histone methylation and acetylation of ADH1 and PDC1 genes, which are jointly upregulated in response to the stressful environment, whereas histones revert to their original state upon stress exposure. Different types of histone modifications also often interact with each other in multiple ways to maintain their status in a somewhat stable manner.

### 3.3. Non-Coding RNA Regulation

DNA methylation and histone modification are regulated at the transcriptional level. In contrast, regulation at the post-transcriptional level is mainly mediated by non-coding RNAs (ncRNAs), a class of molecules that contain genetic material but do not encode proteins, which are produced by the transcription of the genome of an organism.

Many known non-coding RNAs can be classified into linear ncRNAs and circular ncRNAs (circRNAs) based on their different structures. Linear ncRNAs include small ncRNAs such as microRNAs, miRNAs, and long-chain ncRNAs (lncRNAs). ncRNAs also play an essential role in plant growth and development by regulating RNA metabolism, protein modification, and chromatin remodeling at the epigenetic level [27]. miRNAs and lncRNAs have been extensively studied. miRNAs are a class of 20–24nt long and highly conserved small RNAs that cause transcriptional cleavage, degradation, or inhibition of translation of target mRNAs through complementary base pairing with the target site. lncRNA sequences are more than 200nt long and are highly conserved transcripts with complex and diverse mechanisms of action, mainly interacting with large molecules such as DNA, RNA, and proteins but also acting as precursors for small molecules such as miRNA [28,29]. The transcription and shearing of lncRNAs are also regulated by DNA methylation, RNA, and histone modifications. lncRNAs have been found to function as cis- or trans-regulators and modulators of genes through studies in plants over the past two years [30].

For example, snRNAs are small nuclear RNAs that are involved in the processing of mRNA precursors and are the major components of RNA spliceosomes during post-transcriptional processing in eukaryotes; snoRNAs are small nucleolar RNAs that play an important role in ribosomal RNA biosynthesis and also direct the post-transcriptional modification of snRNA, tRNA, and mRNA; and siRNAs are significant members of siRISC and stimulate the silencing of complementary target fragments [31]. With the development of high-throughput sequencing, more and more types and functions of ncRNAs are being identified. In the future, CRISPR/Cas9 will also be widely used to identify ncRNAs.

## 4. Application of CRISPR in Plant Epigenetic Regulation

The emergence of CRISPR/Cas has greatly accelerated the research process of epigenetic modification, which has made a revolutionary breakthrough compared with the previous two generations of gene-editing technology (Figure 1).

### 4.1. Transcriptional Regulation

CRISPRi was first developed and used for selective transcriptional inhibition of target genes in *Escherichia coli* and human cells [37]. CRISPRi provides an efficient and specific genomic targeting platform that does not change the targeted DNA sequence, while transcriptional regulation and the fusion of transcriptional repressor domain (TRD) to dCas9 protein could form chimeric dCas9-TRD protein to strengthen the transcriptional interference [33]. The expression of dCas9 fused 3 × SRDX in common wheat can effectively inhibit the expression of the target gene *TaPDS*, and the inhibition can be transmitted to the next generation [38]. When dCas9 was fused with 3 × D144, 3 × DLS, and 3 × MIX, the expression of multiple endogenous genes located in different parts of leaf cells of the dicotyledonous plant N. *Benthamiana* was significantly downregulated. In addition, these three TRDs, D144, DLS, and MIX, were also successfully verified in monocotyledonous plant wheat to inhibit transcriptional activity and produce an albino phenotype [33]. In the regulation pathway of secondary metabolites, CRISPRi successfully mediated the downregulation of the Cinnamate-4-Hydroxylase (C4H) gene and enhanced flavonoid biosynthesis in *Nicotiana tabacum* [39].

The CRISPR/dCas9 activation (CRISPRa) system can overcome the limitations of traditional gene overexpression methods, and multiple gene activation can be achieved by specifying multiple guide RNAs in one vector [40]. In this system, dCas9 can be fused with different transcription activation domains (TADs) and other transcription activators to mediate the activation of target gene expression [41,42,43]. When the VP64 transcriptional activation domain was fused to the C-terminus of inactivated pcoCas9 and co-expressed with gRNAs targeting multiple target gene promoters, the expression of the protein-coding gene *Anthocyanin Pigment1* (*AtPAP1*) and the non-protein-coding gene *miR319* (a microRNA) were significantly activated [44]. A dCas9-TV system fused with VP128 with up to six copies of the TALE TAD motif was developed through plant cell screening. Compared with dCas9-VP64, it has a wide range of effectiveness and higher activation efficiency for target gene activation in various plant cells [45]. Subsequently, the CRISPR-Act2.0 system was generated by coupling dCas9-VP64 with MS2-VP64 through gRNA2.0. The transcriptional activation efficiency of the CRISPR-Act2.0 system was three to four times higher than that of the first generation dCas9-VP64 system and was used for transcriptional activation in rice and Arabidopsis. In addition, the author also developed and tested multiplexed TALE-activation (mTALE-Act) systems. Two compatible gateway entry vectors, pYPQ121 and pYPQ127B, allow the expression of two TALE-VP64 activators respectively. Therefore, the system can target up to four gene activation sites at the same time. In subsequent tests, simultaneous transcriptional activation of three independent target genes in *Arabidopsis* was successfully achieved [34].

### 4.2. DNA Methylated and Demethylated

CRISPR/dCas can be used to recruit different epigenetic effector domains for specific epigenetic regulation of target sites [35,46]. DNA methylation is established by two de novo DNA methyltransferases (DNMT3a/b), and DNMT1 maintains the methylation of CG sites in mammals [47,48]. Fusion of the dCas9 endonuclease with the DNMT3a catalytic domain (the functional domain) via a flexible Gly4Ser linker generated a CRISPR-Cas9-based tool for targeted CpG methylation. Under the guidance of multiple sgRNAs, the activity of the dCas9-DNMT3a increased the methylation level of the wider region within the promoter of *IL6ST* and *BACH2* sites and reduced their expression levels in human embryonic kidney cell [49]. The fusion of the catalytic domain of human demethylase ten-eleven translocation 1 (TET1cd) with an artificial zinc finger (ZF) can promote the demethylation of the *FWA* promoter in Arabidopsis, upregulate the expression, and produce a heritable late flowering phenotype. At the same time, the author also developed a targeted demethylation system based on CRISPR/dCas9 system that combines TET1cd and an improved SunTag system. SunTag-TET1cd can also demethylate and activate gene expression when targeting *FWA* or *CACTA1* [50]. Domains rearranged methyltransferase (DRM) is a key de novo methyltransferase in plants [51]. Based on the CRISPR-Cas9-SunTag targeting system, the VP64 in the SunTag system was replaced with the tobacco DRM methyltransferase catalytic domain (NtDRMcd) as the methylation effector, which successfully made the Arabidopsis FWA promoter ectopic methylation and early flowering phenotype [35].

### 4.3. Histone Acetylation

Co-expression of CRISPR-Cas9 histone acetyltransferase (HAT), which fuses dCas9 protein with the human acetyltransferase p300, with a single sgRNA catalyzes the acetylation of histone H3 at lysine 27 in the enhancer region, thereby specifically and strongly activating downstream target genes [52]. A similar system effectively increased the H3K27ac level for the flowering time (*FT*) gene. H3K27ac has a slight effect on mRNA level and flowering time [53,54]. DCas9 was fused with an Arabidopsis histone acetyltransferase 1, which improved the abscisic acid-responsive element-binding protein *AREB1/ABF2* gene expression by remodeling chromatin, successfully improving the drought tolerance of *Arabidopsis* [36]. This indicates that dCas^HAT^ system has a good application prospect in plant epigenetic markers.

### 4.4. Non-Coding RNA

At present, the research progress of non-coding RNA is slow in epigenetic modification. The CRISPR/Cas system can be used for lncRNA targeting, knockout, knockdown, overexpression, and imaging [55]. The genome-scale CRISPR-Cas9 activation screening system was developed to systematically explore the loci of long non-coding RNA (lncRNA) and help elucidate its functional characteristics [56]. Subsequently, a new system named CERTIS came into being. It is mediated by CRISPR and is used for the localization and immune interaction verification of endogenous lncRNA [57]. It is suitable for tracking the temporal and spatial changes of lncRNA in living cells, and can also be used to study the interaction mechanism of lncRNA [57]. The catalytic inactivation CasRx (dCasRx) system was developed to splice RNA [58], lncRNAs undergo similar splicing to RNA splicing, so the dCasRx system can perform splicing isomerism engineering on transcripts [55].

### 4.5. Summary and Prospects

Epigenetic modification is widely involved in plant growth and development. With the rapid development of molecular biology techniques such as gene editing, epigenetics has shown great potential in the genetic improvement of important traits of crops. The emergence of epigenetic genome editing technology also provides technical support for applying epigenetic modification in stress resistance genetic improvement. However, the mechanism of epigenetics in plant response to biotic and abiotic stress, vegetative growth, and reproductive growth has not been systematically studied.

Plant epigenetics has broad application prospects in horticultural plants. The characteristics of plant asexual reproduction make epigenetic genome editing easier to maintain and more stable in offspring. Molecular breeding combining gene editing to regulate epigenetics shows great application potential and research value. Epigenetic editing can be used to study the function of plant transcription factors. It can regulate the expression of key enzyme genes in plant metabolic pathways and affect the synthesis of metabolites. It can also optimize the expression of plant functional genes to improve the biological and abiotic resistance of plants. With the continuous development of high-throughput sequencing technology, biological research has entered the era of big data. In the future, plant stress memory research will no longer stay at the level of single omics, but will be transformed into multi-omics joint analysis to systematically explain the regulatory network in stress memory.

## Figures and Tables

**Figure 1 ijms-24-03442-f001:**
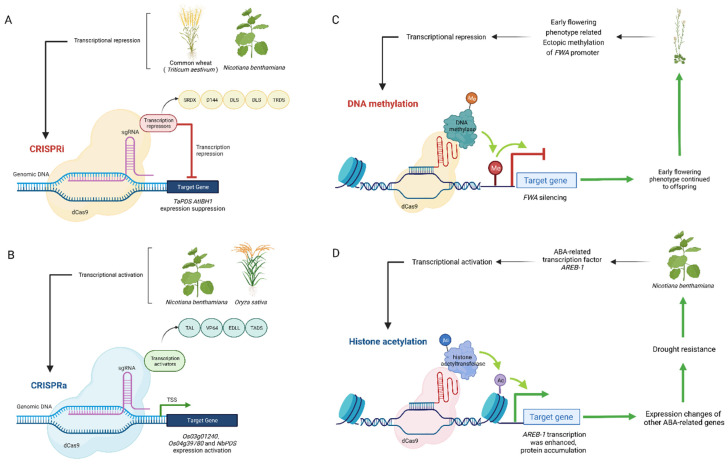
Application of CRISPR in plant epigenetic regulation (by Biorender). (**A**) CRISPRi inhibits the transcription of target DNA by forming a steric hindrance through the targeted binding of dCas9-sgRNA complex to target DNA. Piatek et al. fused the C-terminus of dCas9 to the transcriptional repressor domain of SRDX and successfully suppressed the transcription of endogenous genes in plant cells [32]. In addition, D144, DLS, and MIX were also successfully verified in monocotyledonous plant wheat to inhibit transcriptional activity [33]. (**B**) Activation of specific gene expression can be achieved by fusion of dCas9 with transcriptional regulators such as VP64 and EDLL. The expression of chimeric d Cas9: EDLL and d Cas9: TAD transcriptional activators in *N. benthamiana* can activate the expression of target gene *PDS*, and different transcriptional activators need to target different distances from TSS to obtain the best transcriptional activation level [32]. The dCas9-VP64 coupled with MS2-VP64 via gRNA2.0 constitutes a new system CRISPR-Act2.0, which successfully performed robust and multiple activation of three genes *Os03g01240*, *Os04g39780* and *Os11g35410* in rice [34]. (**C**) The CRISPR-Cas9 SunTag NtDRMcd system was used to methylate the specific site of *FWA* promoter in *Arabidopsis*, which successfully inhibited the expression of *FWA* and caused early flowering phenotype in *Arabidopsis* [35]. (**D**) The dCas9 was fused with an Arabidopsis histone acetyltransferase 1, histone acetylation causes structural changes in chromatin and promotes the assembly of transcription mechanisms, resulting in enhanced gene transcription and subsequent protein accumulation. The AREB1 overexpression promotes an improvement in the physiological performance of the transgenic homozygous plants under drought [36].

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
