# Peer review of "Advances in Plant Epigenome Editing Research and Its Application in Plants"

_ijms, 2023, doi:10.3390/ijms24043442_

Round 1

Reviewer 1 Report

Qi et al. have extensively summarized the knowledge of plant epigenetics and their application to the current state of epigenome editing technology. This is a useful review to gain knowledge about epigenomic technology in the plant field, which has been gradually becoming more successful in recent years, but there are a few corrections that need to be made before publication. The reviewer believe that the quality will be increased if the authors can address the following queries.

<Major concerns>

The citation is inadequately described in the following parts.

-L.159-162

-L.255-256

-L.349-350 (Subsequently, ~~~~~ came into being.)

I think that the references that should be cited in the following parts are incorrect.

-L.198-201 (A typical example is ~~~~ flowering-related genes)

-L.356-358

L.200-201

The authors should specify the gene names that ‘flowering-related genes’ refer to.

L.309

The authors should describe the principles of the mTALE-Act system in more detail.

<Minor concerns>

L.50-51

Since the sentence is a duplicate of lines 47-48, please delete one of the two.

L.203

the nutrient growth -> the vegetative growth?

Author Response

We have made changes in response to your comments, please see the attachment.

Reviewer 2 Report

This report “Advances in plant epigenome editing research and its application in plants” comprehensively reviews the genome editing technologies, types of epigenomes, and the successful application of CRISPR/Cas9 in plant epigenomes. However, this review lacks focus and rigor. The manuscript needs restructuring and substantially revised. The authors must include a figure that summarize the application of CRISPR in plant epigenetic regulation. A point that must imperatively be addressed in 3.1 topic, is how it is how the different types of DNA modifications (DNA methylation, DNA deamination, DNA alkylation, DNA oxidation and Cross-links) are characterized with epigenetic mechanisms. The same goes for item 3.2, for the various modifications of histones (phosphorylation, ubiquitination, methylation, and acetylation), the description given by the authors is very superficial. The authors should critically discuss the existing literature, point out the knowledge gaps, and suggest further research. The manuscript slightly lacks coherence in storyline, and English language also needs careful editing for better readability. The manuscript should be further strengthened by adding some more relevant papers. The literature search is insufficient, only few related research papers in the past five years are cited (54%, approximately), add the latest research results appropriately.

Author Response

We have made changes to the article based on your review comments, including adding incomplete sections of discussion, thoroughly checking references, correcting inappropriate citations, and refining figures. Please see the attachment.
